# Revisiting K-mer Profile for Effective and Scalable Genome Representation Learning

**Abdulkadir Çelikkanat**
Aalborg University
9000 Aalborg, Denmark
abce@cs.aau.dk

**Andres R. Masegosa**
Aalborg University
9000 Aalborg, Denmark
arma@cs.aau.dk

**Thomas D. Nielsen**
Aalborg University
9000 Aalborg, Denmark
tdn@cs.aau.dk

## Abstract

Obtaining effective representations of DNA sequences is crucial for genome analysis. Metagenomic binning, for instance, relies on genome representations to cluster complex mixtures of DNA fragments from biological samples with the aim of determining their microbial compositions. In this paper, we revisit $k$-mer-based representations of genomes and provide a theoretical analysis of their use in representation learning. Based on the analysis, we propose a lightweight and scalable model for performing metagenomic binning at the genome read level, relying only on the $k$-mer compositions of the DNA fragments. We compare the model to recent genome foundation models and demonstrate that while the models are comparable in performance, the proposed model is significantly more effective in terms of scalability, a crucial aspect for performing metagenomic binning of real-world datasets.

## 1 Introduction

Microbes influence all aspects of our environment, including human health and the natural environment surrounding us. However, understanding the full impact of the microbes through the complex microbial communities in which they exist, requires insight into the composition and diversity of these communities [19].

Metagenomics involves the study of microbial communities at the DNA level. However, sequencing a complex microbial sample using current DNA sequencing technologies [27] rarely produces full DNA sequences, but rather a mixture of DNA fragments (called *reads*) of the microbes present in the sample. In order to recover the full microbial genomes, a subsequent binning/clustering step is performed, where individual DNA fragments are clustered together according to their genomic origins. This process is also referred to as *metagenomic binning* [19, 9]. The fragments being clustered during the binning process consist of contiguous DNA sequences (*contigs*) obtained from the reads through a so-called assembly process [29] (contigs are generally longer and less error-prone than the reads).

Metagenomic binning typically involves comparing and clustering DNA fragments using a distance metric in a suitable genome representation space. State-of-the-art methods for metagenomic binning typically rely on representations that include or build on top of the $k$-mer profiles of the contigs [17, 12, 18]. These representations have mostly been studied from an empirical perspective [6], although general theoretical analyses into their representational properties have also been considered [25]. With $k = 4$ (i.e., tetra-nucleotides) a 256-dimensional vector is used to describe the genome, each entry in the vector encoding the frequency of a specific $k$-mer (e.g., ACTG, ATTT) in the genome sequence. The $k$-mer representation of a sequence thus has a fixed size and is independent of sequence length and downstream analysis tasks, providing a computationally efficient representation.

38th Conference on Neural Information Processing Systems (NeurIPS 2024).

A more recent and popular line of research focuses on using approaches inspired by modern Large Language Models (LLMs) to derive more powerful representations of genome fragments. The goal is to replicate the success of LLMs used in natural language processing for genomic data. These models, known as genome foundation models, have seen numerous versions proposed recently [32, 16, 31].

Similarly to popular LLMs, existing genome foundation models utilize next-token prediction or masked-prediction approaches within transformer-based architectures, where the tokens to be predicted are the nucleotides composing the genome fragments. These models, akin to LLMs, enable trainable and contextualized representations that can be defined either in a task-dependent or task-independent manner [32]. According to standardized benchmarks, the embeddings derived from these foundation models have the potential to offer substantial improvements over those based on $k$-mers [32]. However, these embeddings are also computationally far more intensive, reducing their scalability in light of the massive amounts of data generated by modern sequencing tech-

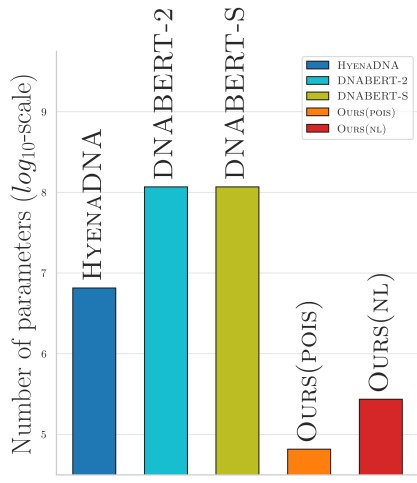

Figure 1: Number of parameters of the different evaluated models ($\log_{10}$-scale).

nologies. For instance, [23] conducted $k$-mer-based metagenomic binning [28, 9] on samples from wastewater treatment plants, encompassing approximately $1.9 \times 10^{12}$ base pairs and resulting in the recovery of over 3700 high to medium quality clusters/metagenome-assembled genomes (MAGs).

In this paper, we demonstrate how $k$-mer-based embeddings of genome fragments provide a scalable and lightweight alternative to genome foundation models. We revisit the theoretical basis of $k$-mers and offer a theoretical characterization of the identifiability of DNA fragments based on their $k$-mer profiles. For non-identifiable fragments, we establish lower and upper bounds on their edit distance using the $l_1$ distance between their respective $k$-mer profiles. These findings offer theoretical justifications for $k$-mer-based genome representations and hold potential significance beyond the scope of this study. Building on these theoretical insights, we propose a simple and lightweight model for learning embeddings of genome fragments using their $k$-mer representations.

We empirically assess the proposed embeddings on metagenomic binning tasks and compare their performance with large state-of-the-art genome foundation models. Our findings indicate that, while both sets of models produce comparable quality in terms of the MAGs recovered, the proposed models require significantly fewer computational resources. Figure 1 demonstrates this by showing the number of parameters in our $k$-mer embedding methods compared to those in the state-of-the-art genome foundation models. The $k$-mer-based embedding approaches involve models with several orders of magnitude fewer parameters.

The main contributions of the paper can be summarized as follows:

- We provide a theoretical analysis of the $k$-mer space, offering insights into why $k$-mers serve as powerful and informative features for genomic tasks.
- We demonstrate that models based on $k$-mers remain viable alternatives to large-scale genome foundation models.
- We show that scalable, lightweight models can provide competitive performance in the metagenomic binning task, highlighting their efficiency in handling complex datasets.

The datasets and implementation of the proposed architectures can be found at the following address: `https://github.com/abdcelikkanat/revisitingkmers`.

## 2   Background and Related Works

When analyzing the microbial content of biological samples (e.g., soil samples or samples from the intestine), we deal with genomic material originating from multiple different species. For such

analyses, the general workflow consists of sequencing the DNA fragments in the sample, producing electrical signals that are subsequently converted into sequences of letters corresponding to the four DNA bases (A, C, T, G). During the sequencing process, the full DNA sequences are fragmented into smaller subsequences (called *reads*), resulting in millions or even billions of reads for each sample. Depending on the sequencing technology being used, reads are typically either classified as short reads (100-150 bases) or long reads (2k-30k bases), and while long reads are generally preferable, they are also more prone to translation errors.

Identifying the microbial composition of a sample involves clustering the DNA fragments according to their genomic origin (a process referred to as *metagenmoic binning*).[1] Clustering the DNA fragments relies not only on a representation of the genome fragments, but typically also includes information about the read coverages in the sample, reflecting the relative abundances of the individual species that make up the sample[2].

For the current paper, the focus is on representations of DNA fragments. Existing works on metagenomic binning typically either rely on predefined features (e.g., [9, 17, 12]) or, more recently, on genome foundation models (e.g., [5, 16, 32]) for representing the DNA fragments. Predefined features for metagenomic binning include the $k$-mer frequencies of the genome (e.g., for $k = 4$, the genome is represented by a 256-dimensional vector), which have shown to exhibit a degree of robustness across different areas of the same genome [4]. These $k$-mer vectors/profiles are either used as direct representations of the DNA fragments as in, e.g., METABAT2[9], VAMP [17], and SEMIBIN2 [18] or is used for learning $k$-mer-based embeddings [15, 21] in the spirit of WORD2VEC [14].

More recently, genome foundation models have also been proposed, which include DNABERT [8], DNABERT-2 [31], HYENADNA [16], and DNABERT-S [32]. These foundation models provide embedding of the raw DNA fragments as represented by the sequences of four nucleotide letters.

In this work, we show that although the contextualized embeddings produced by foundation models provide a strong basis for downstream genome analysis tasks, the complexity of the models also makes them less scalable to the vast amounts of data currently being generated by the state-of-the-art sequencing technologies. Consequently, we consider more lightweight models and explore the use and theoretical basis for models relying on $k$-mer representations of DNA fragments.

# 3  $k$-mer embeddings

As we discussed in the previous section, the metagenomic binning problem aims to cluster the sequences (i.e. reads) according to their respective genomes. For the sake of the argument, we denote with $\ell : \mathcal{R} \to [K]$ the function that maps each read to the (index) ground-truth genome from which the read originates. Here, we use $\mathcal{R}$ to denote the set of reads where each read is supposed to originate from one of the $K$ genomes.

Performing clustering directly on the reads has been shown to provide very poor performance [11], mainly because reads lie in a complex manifold in a very high dimensional space. A well-tested approach is to instead *embed* each of the objects in a lower dimensional space, known as the *embedding or latent space*, whose structure is much closer to Euclidean space, where clustering is far simpler to perform. More formally, we characterize the problem as follows:

**Problem Definition.** Let $\mathcal{R} \subset \Sigma^+$ be a finite set of reads with a genome mapping function $\ell$ where $\Sigma = \{A, C, T, G\}$. For a given threshold value $\gamma \in \mathbb{R}^+$, the objective is to learn an embedding function $\mathcal{E} : \mathcal{R} \to \mathbb{R}^d$ that embeds reads into a low-dimensional metric space $(\mathsf{X}, d_\mathsf{X})$, usually a Euclidean space, such that $d_\mathsf{X}(\mathcal{E}(\mathbf{r}), \mathcal{E}(\mathbf{q})) \leq \gamma$ if and only if $\ell(\mathbf{r}) = \ell(\mathbf{q})$ for all reads $\mathbf{r}, \mathbf{q} \in \mathcal{R}$ where $d \ll |\mathcal{R}|$.

Once we establish an embedding function like the one described earlier, it becomes, in principle, straightforward to employ a simple clustering algorithm to bin or group the reads. However, for this purpose, the embedding function $\mathcal{E}$ must be capable of distinguishing the intrinsic features of the various genomes, producing similar latent representations for reads that originate from the same genomes. In the following subsection, we will discuss novel theoretical arguments that help to better understand why $k$-mer profiles are a powerful representation of reads.

---

[1]In practice, the reads are not clustered directly. Instead overlapping reads are first combined into so-called contigs through an assembly process [10].

## 3.1 k-mers are a powerful representation of reads

The use of $k$-mer profiles to represent reads helps overcome several challenges encountered during the clustering or binning of reads. These challenges include (i) the variation in read lengths in practical scenarios, such as in genome sequencing where read lengths can differ significantly, (ii) ambiguity in read direction, which is a result of reads lacking inherent directionality, complicating the analysis, and (iii) the equivalence of a DNA sequence to its complementary sequence because DNA is composed of two complementary strands, meaning the information from one strand corresponds to that from its complementary strand.

An initial important question concerns the *identifiability* of reads, which examines to what extent different reads share the same $k$-mer profiles. We will call the reads that can be uniquely reconstructed from their given $k$-mer profile as **identifiable** and this is crucial because if many reads possess the same $k$-mer profile, they will invariably be grouped into the same cluster, potentially leading to the loss of significant information. In this context, Ukkonen et al. [25] conjectured in their study on the string matching problem that two sequences sharing the same $k$-mer profile could be transformed into one another through two specific operations, and this claim was later proved by Pevzner [20]. However, these specified operations fail to consider cases where sequences include repeated overlapping occurrences of the same $k$-mer.

In this regard, in Theorem 3.1, we demonstrate that a read can be perfectly reconstructed from its $k$-mer profile under certain conditions, which become less restrictive with larger $k$ values. In the following theorem, we use $r_i \cdots r_j$, with $i < j$, to denote any subsequence of a read $r$ that includes consecutive nucleotides from position $i$ to position $j$.

**Theorem 3.1.** *Let* $\mathbf{r}$ *be a read of length* $\ell$. *There exists no other distinct read having the same $k$-mer profile if and only if it does not satisfy any of the following conditions:*

1. $r_1 \cdots r_{k-1} = r_{\ell-k-2} \cdots r_\ell$ *and* $r_i \neq r_1$ *for some* $1 < i < \ell - k - 2$.

2. $r_i \cdots r_{i+k-2} = r_j \cdots r_{j+k-2}$ *and* $r_g \cdots r_{g+k-2} = r_h \cdots r_{h+k-2}$ *for some indices* $1 \leq i < g < j < h \leq \ell - k + 2$ *where* $r_{i+k-1} \cdots r_{g-1} \neq r_{j+k-1} \cdots r_{h-1}$.

3. $r_i \cdots r_{i+k-2} = r_j \cdots r_{j+k-2} = r_h \cdots r_{h+k-2}$ *for some indices* $1 \leq i < j < h \leq \ell - k + 2$ *where* $r_{i+k-1} \cdots r_{j-1} \neq r_{j+k-1} \cdots r_{h-1}$.

*Proof.* The proof is omitted due to the limitation on the number of pages. Please see the appendix. $\square$

It is important to note that it is always possible to find an optimal $k^*$ value that ensures that each read does not meet any of the previously mentioned conditions and, in consequence, becomes identifiable. Clearly, when $k$ is equal to the length of the reads, all reads become identifiable. However, as $k$ increases, the $k$-mer profiles become more prone to errors, and it will be harder to identify the underlying patterns in the reads (in the extreme case scenario where $k$ equals the length of the reads, each read will correspond to a unique $k$-mer). Thus, using large values of $k$ is impractical. However, the results show that even with smaller $k$ values, identifiability could still be potentially achieved if not by all but by some of the reads. We hypothesize that the effectiveness of $k$-mer representations stems, in part, from this fact.

Another perspective to consider is the extent to which two similar reads share a similar $k$-mer profile. The identifiability approach from the previous result addresses this issue by establishing that if two reads have identical profiles, then the reads themselves are identical. However, identifiability is limited by the conditions outlined in Theorem 3.1, which may not always be met. The observation given in Proposition 3.2 provides a broader and more applicable finding.

It demonstrates that if two $k$-mers are similar according to the $l_1$ distance, then the corresponding reads are also similar according to the Hamming distance in the read space. It is important to note that both the $l_1$ and Hamming distances are natural measures for evaluating similarity in this context. Technically speaking, we show that the $l_1$ distance between $k$-mer profiles can be upper and lower bounded by the Hamming distance between the corresponding identifiable reads, so they are Lipschitz equivalent spaces. In the next result, $c : \mathcal{R} \times \Sigma^k \to \mathbb{N}$ denotes the function showing the number of occurrences of a given $k$-mer in a given read. For simplicity, we use the notation, $c_{\mathbf{r}}$, to indicate the $k$-mer profile vector of a read $r$ whose entries are ordered lexicographically. In other words, $c_{\mathbf{r}}(\mathbf{x})$ is the number of appearances of $\mathbf{x} \in \Sigma^k$ in read $\mathbf{r} \in \mathcal{R}$.

**Proposition 3.2.** *Let $M_1 = (\aleph_\ell, d_{\mathcal{H}})$ and $M_2 = (\mathbb{N}^{|\Sigma^k|}, \|\cdot\|_1)$ be the metric spaces denoting the set of identifiable reads and their corresponding $k$-mer profiles equipped with edit and $\ell_1$ distances, respectively. The $k$-mer profile function, $c : M_1 \to M_2$, mapping given any read, $\mathbf{r}$, to its corresponding $k$-mer profile, $c_{\mathbf{r}} := c(\mathbf{r})$, is a Lipschitz equivalence, i.e. it satisfies*

$$\forall \mathbf{r}, \mathbf{q} \in \Sigma^\ell \quad \alpha_l d_{\mathcal{H}}(\mathbf{r}, \mathbf{q}) \le \|c_{\mathbf{r}} - c_{\mathbf{q}}\|_1 \le \alpha_u d_{\mathcal{H}}(\mathbf{r}, \mathbf{q}) \tag{1}$$

*for $\alpha_l = 1/\ell$ and $\alpha_u = k|\Sigma|^k$, so $M_1$ and $M_2$ are Lipschitz equivalent.*

*Proof.* The proof is omitted due to the limitation on the number of pages. Please see the appendix. ☐

Unfortunately, when one runs a clustering algorithm directly on top of the *raw* $k$-mer profiles, the results are not competitive as demonstrated in numerous studies, see, e.g., [31]. Instead, we will consider methods for learning embeddings of $k$-mer profiles and, more generally, genome sequences.

### 3.2 Linear read embeddings

Below we consider two simple genome sequence models defining the embedding of a read as a linear combination of the $k$-mer representations. The first model ($k$-mer profile) does not involve any learning procedure and it is simply defined by the $k$-mer counts of the read, whereas the second model (Poisson model) expands on the $k$-mer profile model by explicitly representing the dependencies between $k$-mers. Both models will serve as baselines for more complex models.

$k$**-mer profile:** First, consider the definition of $k$-mer profiles:

$$\mathcal{E}_{\text{KMER}}(\mathbf{r}) := \sum_{\mathbf{x} \in \Sigma^k} c_{\mathbf{r}}(\mathbf{x}) \mathbf{z}_{\mathbf{x}} \tag{2}$$

where $\mathbf{z}_{\mathbf{x}}$ represents the canonical basis vector for the $k$-mer $\mathbf{x} \in \Sigma^k$, i.e. $(\mathbf{z}_{\mathbf{x}} \in \{(u_1, \dots, u_{|\Sigma^k|}) \in \{0,1\}^{|\Sigma^k|} : \sum_i u_i = 1\}$ ).

As previously mentioned, our primary objective is to represent reads in a lower-dimensional space, where their relative positions in a latent space reflect their underlying similarities. By Equation 2, the $k$-mer profile of a read can also be considered as its embedding vector, but its entries are not independent. In other words, if we change one letter in a read, it can affect at most $k$ different $k$-mer counts. In this regard, it might be possible to learn better representations than simple $k$-mer profiles.

**Poisson model:** For a given read $\mathbf{r} := (r_1 \dots r_\ell)$, it is easy to see that the consecutive $k$-mers in the read are not independent. For instance, the $k$-mers located at position indices $i$ and $i + 1$, (i.e. $\mathbf{x} := r_i \cdots r_{i+k-1}$ and $\mathbf{y} := r_{i+1} \cdots r_{i+k}$), share the same substring $r_{i+1} \cdots r_{i+k-1}$, which highlights the inherent dependencies between $k$-mers. Hence, we will learn the representation, $\mathbf{z}_{\mathbf{x}}$, of each $k$-mer, $\mathbf{x}$, instead of using the canonical basis vectors. We propose to model the co-occurrence frequency of specific $k$-mer pairs within a fixed window size, $\omega$, using the Poisson distribution. In other words, $o_{\mathbf{x},\mathbf{y}} \sim Pois(\lambda_{\mathbf{x},\mathbf{y}})$, where $o_{\mathbf{x},\mathbf{y}}$ indicates the number of co-appearances of $k$-mers, $\mathbf{x} := r_i \cdots r_{i+k-1}$ and $\mathbf{y} := r_j \cdots r_{j+k-1}$ for $|i - j| \le \omega$.

In order to achieve close representations for highly correlated $k$-mers, and distant embeddings for the dissimilar ones, we define the rate of the Poisson distribution as follows:

$$\lambda_{\mathbf{x},\mathbf{y}} := \exp\left(-\|\mathbf{z}_{\mathbf{x}} - \mathbf{z}_{\mathbf{y}}\|^2\right). \tag{3}$$

The loss function used for training the embeddings with respect to the reads is simply defined in terms of the Poisson distribution model of the $k$-mer pairs:

$$\mathcal{L}_{\text{POIS}}\left(\{o_{\mathbf{x},\mathbf{y}}\}_{\mathbf{x},\mathbf{y} \in \Sigma^k} | \{\mathbf{z}_{\mathbf{x}}\}_{\mathbf{x} \in \Sigma^k}\right) := \frac{1}{2} \sum_{\mathbf{x} \in \Sigma^k} \sum_{\mathbf{y} \neq \mathbf{x} \in \Sigma^k} o_{\mathbf{x},\mathbf{y}} \|\mathbf{z}_{\mathbf{x}} - \mathbf{z}_{\mathbf{y}}\|^2 + \exp\left(-\|\mathbf{z}_{\mathbf{x}} - \mathbf{z}_{\mathbf{y}}\|^2\right) \tag{4}$$

where $o_{\mathbf{x},\mathbf{y}}$ is the average number of co-occurrences of $k$-mers $\mathbf{x}$ and $\mathbf{y}$ per read within a window of size $\omega$ in the dataset. Once we have obtained the $k$-mer embeddings, we can obtain the read embeddings by combining the $k$-mer embeddings with their respective occurrence counts, as given in Equation 2. More specifically, the embedding of read, $\mathbf{r}$, is given by

$$\mathcal{E}_{\text{POIS}}(\mathbf{r}) = \frac{1}{\sum_{\mathbf{x} \in \Sigma^k} c_{\mathbf{r}}(\mathbf{x})} \sum_{\mathbf{x} \in \Sigma^k} c_{\mathbf{r}}(\mathbf{x}) \mathbf{z}_{\mathbf{x}} \tag{5}$$

where $\{\mathbf{z}_{\mathbf{x}}\}_{\mathbf{x} \in \Sigma^k}$ corresponds to the learned $k$-mer embeddings of Eq. 4.

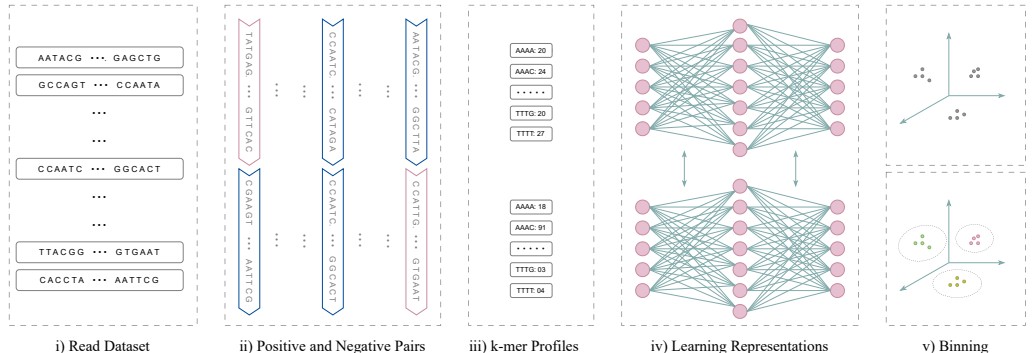

| i) Read Dataset | ii) Positive and Negative Pairs | iii) k-mer Profiles | iv) Learning Representations | v) Binning |

Figure 2: Illustration of the non-linear $k$-mer embedding approach described in Section 3.3.

## 3.3    Non-linear read embeddings

In the experimental section, we will show that the linear $\mathbf{k}$-mer embeddings described previously outperform raw $\mathbf{k}$-mer profiles in metagenomic binning tasks. However, the literature on machine learning frequently emphasizes that non-linear embeddings typically provide superior results when the data, such as genomic sequences in this context, exist in complex high-dimensional manifold spaces [33].

We design a simple and efficient neural network architecture utilizing self-supervised contrastive learning. Our model consists of two linear layers, and the first one takes $k$-mer profile features as input, projecting them into a $512$-dimensional space. A sigmoid activation function is applied, followed by a batch normalization operation and a dropout with a ratio of $0.2$. The second layer maps them into the final $256$-dimensional embedding space.

Our approach is inspired by previous methods in the context of metagenomic binning at the contig level [18]. The primary aim here is not to introduce a novel methodology for learning embeddings but to demonstrate that simple non-linear embeddings, built upon $k$-mer representations, can match the effectiveness of existing state-of-the-art genomic foundation models for metagenomic binning tasks while being significantly more scalable.

The methodology is graphically depicted in Figure 2. Initially, we create positive and negative pairs of read segments using the dataset provided. To create a positive pair, we split an existing read into two equal-sized segments. Conversely, a negative pair is formed by combining two segments originating from the splitting of two distinct reads chosen at random. Subsequently, for each pair, we calculate the $k$-mer profile for the two segments in the pair. These two $k$-mer profiles are then input into the same neural network, which maps them to two vectors within the embedding space. The loss function evaluates the quality of these two embeddings; it penalizes distant embeddings in positive pairs and close embeddings in negative pairs. The learning process involves optimizing the neural network's weights to minimize this loss function across numerous positive and negative samples.

Let $\mathcal{R} = \{\mathbf{r}_i\}_{i=1}^N$ be the set of reads and $\mathbf{r}_i^l$ and $\mathbf{r}_i^r$ are the segments indicating the left and right halves of read $\mathbf{r}_i \in \mathcal{R}$. As described above, we use these segments to construct the positive and negative pair samples required for the self-supervised contrastive learning procedure. Let $\{(\mathbf{r}_i^*, \mathbf{r}_j^*, y_{ij})\}_{(i,j)\in\mathcal{I}}$ be the set of triplets where $\mathcal{I} \subseteq [N] \times [N]$ is the index set for the pairs, and the symbol, $y_{ij} \in \{0, 1\}$ is used to denote whether $(\mathbf{r}_i^*, \mathbf{r}_j^*)$ is a positive sample (i.e. $+1$), or negative (i.e. $0$). The positive label means that the pair corresponds to the left and right segments of the same read, while a negative label indicates they are segments from different reads. Then, we define the loss function as follows:

$$\mathcal{L}_{\text{NL}}\Big(\{y_{ij}\}_{(i,j)\in\mathcal{I}}|\Omega\Big) := -\frac{1}{|\mathcal{I}|}\sum_{(i,j)\in\mathcal{I}} y_{ij}\log p_{ij} + (1 - y_{ij})\log(1 - p_{ij}) \tag{6}$$

where $\Omega$ indicates the all trainable weights of the neural network. Here, the success probability, $p_{ij}$, corresponds to the case of a positive pair, occurring when $y_{ij} = +1$, and computed as:

$$p_{ij} = \exp\left(-\|\mathcal{E}_{\text{NL}}(\mathbf{r}_i) - \mathcal{E}_{\text{NL}}(\mathbf{r}_j)\|^2\right)$$

where $\mathcal{E}_{\text{NL}}(\mathbf{r}_i^*)$ denotes the output embedding of the neural network for segment $\mathbf{r}_i^*$.

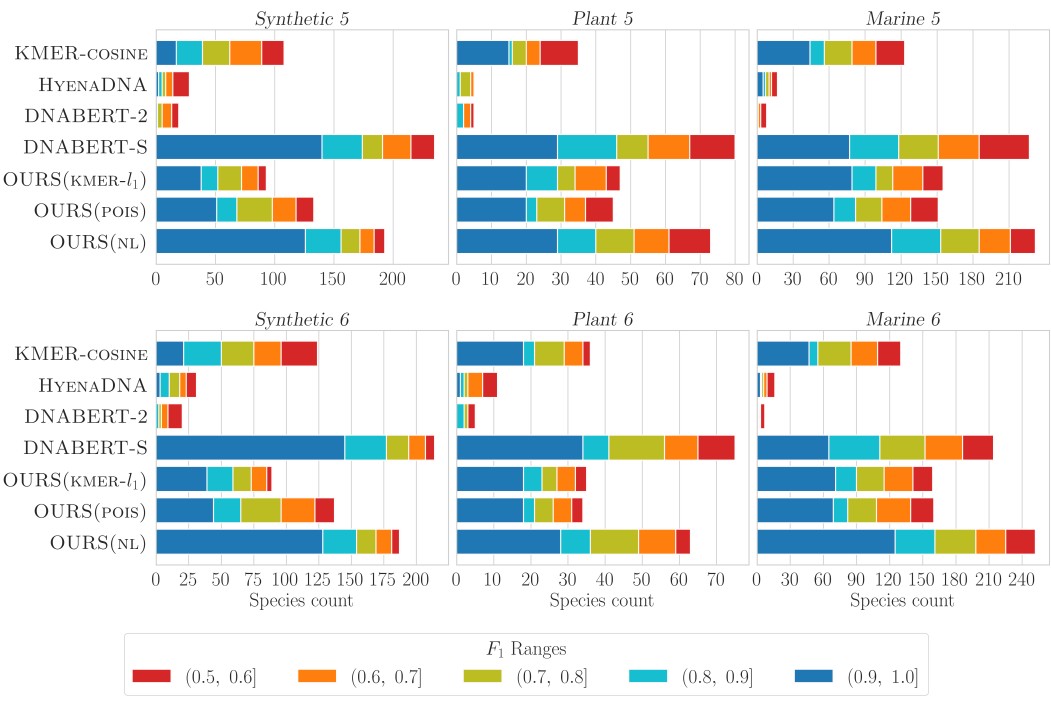

Figure 3: Evaluation of the models on multiple datasets for the metagenomic binning task. Each color represents the number of clusters within a specific $F_1$ score range, and the number of clusters with the highest quality is highlighted in dark blue.

Given a large number of distinct genomes (i.e., clusters), it is highly likely that *negative* pairs consist of segments from different genomes, as they originate from different reads. Conversely, *positive* pairs are composed of segments from the same genome, as they originate from the same read. Consequently, the neural network's embedding function will learn to produce similar embeddings for $k$-mer profiles belonging to the same genome and different embeddings for $k$-mer profiles belonging to different genomes. As demonstrated in the next section, this straightforward approach built on $k$-mer profiles competes effectively with state-of-the-art genomic foundation models, which involve neural networks with several orders of magnitude more parameters.

# 4 Experiments

In this section, we will provide the details regarding the experiments, datasets, and baseline approaches that we consider to assess the performance of the proposed linear and non-linear models.

**Datasets**. We utilize the same publicly available datasets used to benchmark the genome foundation models [32]. The training set consists of 2 million pairs of non-overlapping DNA sequences, each $10,000$ bases in length, constructed by sampling from the dataset, including $17,636$ viral, $5,011$ fungal, and $6,402$ distinct bacterial genomes from GenBank [3]. For model evaluation, we use six datasets derived from the CAMI2 challenge data [13], representing marine and plant-associated environments and including fungal genomes. These datasets propose realistic and complex reads, making them one of the most utilized benchmarks for the metagenomics binning task. Additionally, the synthetic datasets consist of randomly sampled sequences from the fungi and viral reference genomes, excluding any samples from the training data [32].

**Baselines and proposed models.** For our experiments, we have employed the recent genome foundation models to assess the performance of our approach. (i) KMER is one of the most widely used baselines, and it also serves as a fundamental component of our model. The DNA sequences are represented as $4$-mer profiles *Tetranucleotide Frequencies* given in Eq. 2 so each read is represented by a 256-dimensional vector. (ii) HYENADNA [16] is a genome foundation model, HYENADNA, pre-trained on the Human Reference Genome [7] with context lengths up to $10^6$ tokens at single

nucleotide resolution. (iii) DNABERT-2 [31] is another foundation model pre-trained on multi-species genomes, and it proposes Byte Pair Encoding (BPE) for DNA language modeling to address the computational inefficiencies related to $k$-mer tokenization. It also incorporates various techniques, such as Attention with Linear Biases (ALiBi) and Low-Rank Adaptation (LoRA), to address the limitations of current DNA language models. (iv) DNEBERT-S [32] aims to generate effective species-aware DNA representations and relies on the proposed Manifold Instance Mixup (MI-Mix) and Curriculum Contrastive Learning approaches. It relies on the pre-trained DNABERT-2 model as the initial point of contrastive training. (v) OURS(KMER-$l_1$) is similar to the KMER approach presented in (i), but it uses $l_1$ distance instead of the cosine-similarity distance so we will also name the first one as OURS(KMER-COSINE) to distinguish both models. (vii) OURS(POIS) refers to the Poisson approach described in Section 3.2. (vi) OURS(NL) refers to the non-linear $k$-mer embedding approach based on self-supervised constrastive learning described in Section 3.3.

**Parameter Settings.** Our proposed models were trained on a cluster equipped with various NVIDIA GPU models. For the optimization of our models, we employed the Adam optimizer with a learning rate of $10^{-3}$. We used smaller subsets of the dataset for training our models, and we sampled $10^4$ reads for OURS(POIS) and $10^6$ sequences for OURS(NL). The OURS(NL) model was trained for 300 epochs with a mini-batch size of $10^4$, while OURS(POIS) was trained for 1000 epochs using full-batch updates and window size of $4$. Since we incorporated the contrastive learning strategy for OURS(NL), we randomly sampled 200 read halves to form the negative instances for each positive sample (one half of a read). We have set $k = 4$ and the final embedding dimension to 256 for our all models. Following the experimental set-up of [32], we used the publicly available pre-trained versions of the baseline genome foundation models in the Hugging Face platform.

## 4.1 Metagenomics Binning Task

We assess the performance of the models based on the number of detected species/clusters and these clusters are classified into five different quality levels in terms of their $F_1$ scores. Our methods and each baseline approach generate embeddings for the reads, and we cluster these read representations by following the work [32] with the modified K-Medoid algorithm. We refer readers to the cited work for further details on this approach. In line with standard practices in the field, we use cosine similarity for the baseline methods to measure the similarity between read embeddings required for the K-Medoid algorithm. For OURS(KMER-$l_1$), we employed the $\exp(-dist(\cdot, \cdot))$ function with $l_1$ distance, while Euclidean distance was used for OURS(POIS) and OURS(NL).

Figure 3 highlights the number of detected bins/clusters across different quality levels. For instance, the number of bins/species whose $F_1$ scores above $0.9$ is represented in dark blue while those with $F_1$ scores between $0.5$ and $0.6$ are highlighted in red. Since we observe that the deviation in the number of detected species in most cases changes between at most $\pm 5$, the error bars were not included.

The results underscore the effectiveness of $k$-mer features, which are central to our paper's focus. More specifically, as it was observed in the study [32], $k$-mer feature vectors (i.e., KMER-COSINE) surpass some genome foundation models like HYENADNA and DNABERT-2. Furthermore, we highlight the importance of choosing an appropriate similarity measure by showcasing the performance of the OURS(KMER-$l_1$) variant. We note that Proposition 3.2 establishes a link between $k$-mer space equipped with $l_1$ distance and read space with edit or Hamming distance. Thus, leveraging an appropriate metric in defining the similarities between reads in the binning stage also contributes to the performance improvement of OURS(KMER-$l_1$).

Despite the OURS(NL) model's lightweight design compared to recent genome foundation architectures, our non-linear embeddings effectively identify both high and low-quality bins. In the metagenomics binning task, for practical purposes, the high-quality bins are typically prioritized (depicted in dark blue), and our model demonstrates comparable performance to the DNABERT-S model on both *Synthetic* and *Plant* datasets, and it also outperforms DNABERT-S on the *Marine* dataset. Overall, our findings emphasize the efficacy of lightweight non-linear embeddings built on top of $k$-mer profiles.

## 4.2 Ablation Study

We conduct a series of ablation studies in order to gain deeper insights into different components of the proposed architectures. We will start first by examining the impact of the parameter $k$ which

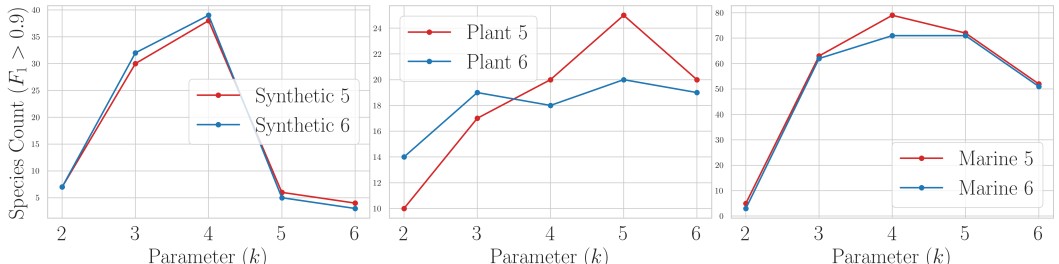

Figure 4: Influence of parameter $k$ on the OURS(KMER-$l_1$) model across different datasets.

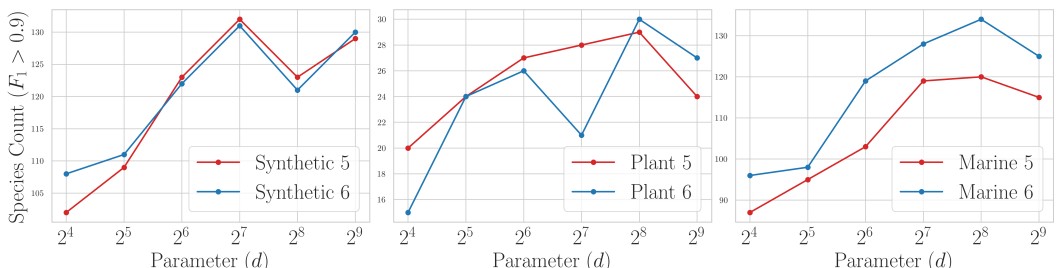

Figure 5: Influence of dimension size $(d)$ on the OURS(NL) model across different datasets.

is used to define the $k$-mer profiles. It plays a fundamental role in the paper because all the model variants that we introduced rely on it. As also explained in Section 3, the selection of $k$ also influences the identifiability of reads. In this regard, we evaluate the performance of the OURS(KMER-$l_1$) model across various $k$ values. Figure 4 shows the number of high-quality bins (i.e., $F_1 > 0.9$) for different $k$ settings and reveals that the optimal performance is generally achieved when $k$ is set to 4 on different datasets. It also holds true across our different architectures, so we have set $k = 4$ in our models.

We also examine the influence of the output dimension size on the performance for the OURS(NL) model. We employed a very shallow architecture as described in Section 3.3, so the output dimension size plays a vital role in the computational cost and performance. As can be observed in Figure 5, the increase in dimension size also positively contributes to the number of detected high-quality bins, and the performance mostly saturates after $2^7$. In order to have a fair comparison with the baselines, we also used $2^8$ as an output dimension size in our architectures.

For the OURS(NL) architecture, it is very natural to use Bernoulli distribution to model whether a given pair of reads belongs to the same genome or not. However, different organisms might share similar regions in their genetic codes; therefore, assuming binary interactions might not suit well in every situation. In this regard, we rewrite Eq. 6 by Poisson distribution that we also employed for the OURS(POIS) architecture as follows:

$$\mathcal{L}_{\text{NL}}^{poisson} := \frac{1}{|\mathcal{I}|} \sum_{(i,j) \in \mathcal{I}} -y_{ij} \log \lambda_{ij} + \lambda_{ij}$$

where $\lambda_{ij} := \exp(-\|\mathcal{E}_{\text{NL}}(\mathbf{r}_i) - \mathcal{E}_{\text{NL}}(\mathbf{r}_j)\|^2)$ and $y_{ij} \in \{0, 1\}$ denotes if reads $i$ and $j$ belong to the same genome or not. We also test the loss function proposed by [18], which we refer as *hinge loss*:

$$\mathcal{L}_{\text{NL}}^{hinge} := \frac{1}{|\mathcal{I}|} \sum_{(i,j) \in \mathcal{I}} y_{ij} \|\mathcal{E}_{\text{NL}}(\mathbf{r}_i) - \mathcal{E}_{\text{NL}}(\mathbf{r}_j)\|^2 + (1 - y_{ij}) \max\{0, 1 - \|\mathcal{E}_{\text{NL}}(\mathbf{r}_i) - \mathcal{E}_{\text{NL}}(\mathbf{r}_j)\|\}^2$$

Table 1 outlines the number of high-quality bins, and the Bernoulli and Poisson loss functions show comparable performances. On the other hand, we observe that it is possible to obtain even higher scores using the Hinge loss. However, it fails to have a probabilistic interpretation.

Table 1: Comparison of loss functions by the number of clusters (with $F_1 > 0.9$).

|  | Synthetic 5 | Synthetic 6 | Plant 5 | Plant 6 | Marine 5 | Marine 6 |
|---|---|---|---|---|---|---|
| Bernoulli | 123 | 121 | 29 | 30 | 120 | 134 |
| Poisson | 128 | 123 | 30 | 28 | 121 | 128 |
| Hinge | 135 | 131 | 40 | 32 | 131 | 140 |

## 5 Conclusion

In this study we have shown the efficacy of non-linear $k$-mer-based embeddings for metagenomic binning tasks, providing a compelling alternative to more recently proposed complex genome foundation models. Our work revisits and expands the theoretical framework surrounding $k$-mers, specifically addressing the identifiability of DNA fragments through their $k$-mer profiles and establishing new bounds on the distances defining relevant metric spaces. This theoretical insight not only reinforces the validity of using $k$-mer-based approaches for genome representation but also highlights their broader applicability in genomic research, such as taxonomic profiling/classification [22, 26] and phylogenetic analysis [30].

The lightweight model being proposed in the paper, grounded in these theoretical principles, shows considerable promise in the field of metagenomic binning. It achieves a performance comparable to that of state-of-the-art genome foundation models while requiring orders of magnitude fewer computational resources. This feature is especially important for large-scale genomic analyses that are currently driven by recent advances in sequencing technologies and where computational efficiency is paramount.

**Limitations and Societal Impact**

The ability to scale up metagenomic binning holds the potential for broader societal impacts through an improved understanding of the diversity and function of the microbial communities that influence our health and environment. Insights into these communities can also play an essential role in achieving the sustainability goals [24, 1], in particular good health and well-being (SDG-3), life below water (SDG-14), and life on land (SDG-15), to name a few. The current study is partly limited by the experimental setup, which is only based on synthetically generated genomic data following the experimental setup of similar studies such as [31, 32]. While the discussions about computational resources are not expected to be significantly affected by the type of data being used, the evaluations and comparisons of the quality of the recovered genomes most likely will. For instance, samples containing genomes of closely related species or strains of the same species will generally be more difficult to separate into distinct clusters. As part of future work, we plan to pursue a more rigorous analysis of the binning quality using long-read sequences from real-world data.

## Acknowledgements

We would like to thank the reviewers for their constructive feedback and insightful comments. We would also like to thank Mads Albertsen, Katja Hose and all members of Albertsen Lab at Aalborg University for the valuable and fruitful discussions. This work was supported by a research grant (VIL50093) from VILLUM FONDEN.

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

# A  Appendix

## A.1  Theoretical Analysis

**Lemma A.1.** *Let* $\mathbf{r} = r_1 \cdots r_\ell$ *be a read satisfying* $c_{\mathbf{r}}(\mathbf{x}_i) = 1$ *for its every* $(k-1)$-*mer,* $\mathbf{x}_i = r_i \cdots r_{i+k-2}$ *where* $1 \le i \le i^*$ *for some* $i^* \in \{1, \ldots, \ell - k + 2\}$. *If there exists a read* $\mathbf{q} = q_1 \cdots q_\ell$ *having the same* $k$-*mer profile as* $\mathbf{r}$, *then* $r_i = q_i$ *for all* $1 \le i \le i^* + k - 2$.

*Proof.* Let $\mathbf{r}$ be a read where $(k-1)$-mers, $\mathbf{x}_i := r_i \cdots r_{i+k-2}$, appears only once in $\mathbf{r}$ for all $1 \le i \le i^*$. As a basis step, we first show that the initial $k$-mer of $\mathbf{q}$ must be the same as that of $\mathbf{r}$, i.e., $q_1 \cdots q_k = r_1 \cdots r_k$. Suppose this is not the case, then there must exist an index $1 \le j \le \ell - k + 1$ such that $q_j \cdots q_{j+k-1} = r_1 \cdots r_k$ since they have identical $k$-mer profiles by initial assumption. Note that $j \ne 1$; otherwise, the reads would share the same prefixes. Therefore, $q_{j-1} \ldots q_{j+k-2}$ is another $k$-mer of $\mathbf{q}$, and it must also be present in $\mathbf{r}$ because they share the same $k$-mer profile. Since $r_1 \cdots r_{k-1} = q_j \cdots q_{j+k-2}$, k-mer $q_{j-1} \cdots q_{j+k-2}$ must be the prefix of $\mathbf{r}$; otherwise $(k-1)$-mer $q_j \ldots q_{j+k-2}$ (i.e. $r_1 \ldots r_{k-1}$) appears in $\mathbf{r}$ more than once and it violates the assumption that $c_{\mathbf{r}}(\mathbf{x}_i) = 1$ for all $\mathbf{x}_i = r_i \cdots r_{i+k-2}$ where $i \in \{1, \ldots, i^*\}$. However, if $q_{j-1} \cdots q_{j+k-2}$ is the prefix of $\mathbf{r}$, then it enforces that $q_{j-1} = q_j = \cdots = q_{j+k-2} = q_{j+k-1}$ and a $(k-1)$-mer occurs more than once in the substring $r_1 \cdots r_{i^*}$ so reads $\mathbf{r}$ and $\mathbf{q}$ must have the same initial $k$-mer.

For the inductive step, suppose that reads $\mathbf{r}$ and $\mathbf{q}$ agree up to position index $j$ where $k \le j < i^* - k - 2$, i.e., $r_i = q_i$ for all $1 \le i \le j$. We will show that $r_{j+1}$ and $q_{j+1}$ are also the same. Since $j < i^* - k + 2$, substring $q_{j-k+2} \ldots q_{j+1}$ is also a $k$-mer of $\mathbf{q}$. Since reads have the same $k$-mer profiles, $\mathbf{r}$ also includes it. By the inductive hypothesis, we have $q_{j-k+2} \cdots q_j = r_{j-k+2} \cdots r_j$, and we know $(k-1)$-mer $r_{j-k+2} \cdots r_j$ occurs only once so it implies that $q_{j-k+2} \cdots q_{j+1} = r_{j-k+2} \cdots r_{j+1}$, i.e. $r_{j+1} = q_{j+1}$. In conclusion, we have $r_i = q_i$ for all $1 \le i \le i^* - k + 2$.  $\square$

**Corollary A.2.** *If the* $(k-1)$-*mer profile of a read contains at most one occurrence for each of its* $(k-1)$-*mers, then there is no other read having the same* $k$-*mer profile.*

*Proof.* Let $\mathbf{r} = r_1 \cdots r_\ell$ and $\mathbf{q} = q_1 \cdots q_\ell$ be reads having the same $k$-mer profile and each of their $(k-1)$-mers appear only once. By Lemma A.1, we have $r_i = q_i$ for all $1 \le i \le \ell$ since $c_{\mathbf{r}}(\mathbf{x}_i) = 1$ for all $(k-1)$-mers $\mathbf{x}_i = r_i \ldots r_{i+k-2}$ where $0 \le i \le \ell - k + 1$  $\square$

**Theorem A.3.** *Let* $\mathbf{r}$ *be a read of length* $\ell$. *There exists no other distinct read having the same* $k$-*mer profile if and only if it does not satisfy any of the following conditions:*

1. $r_1 \cdots r_{k-1} = r_{\ell-k-2} \cdots r_\ell$ *and* $r_i \ne r_1$ *for some* $1 < i < \ell - k - 2$.

2. $r_i \cdots r_{i+k-2} = r_j \cdots r_{j+k-2}$ *and* $r_g \cdots r_{g+k-2} = r_h \cdots r_{h+k-2}$ *for some indices* $1 \le i < g < j < h \le \ell - k + 2$ *where* $r_{i+k-1} \cdots r_{g-1} \ne r_{j+k-1} \cdots r_{h-1}$.

3. $r_i \cdots r_{i+k-2} = r_j \cdots r_{j+k-2} = r_h \cdots r_{h+k-2}$ *for some indices* $1 \le i < j < h \le \ell - k + 2$ *where* $r_{i+k-1} \cdots r_{j-1} \ne r_{j+k-1} \cdots r_{h-1}$.

*Proof.* Let $\mathbf{r} = r_1 \cdots r_\ell$ be a read of length $\ell$, and we will first show that if a read satisfies one of these conditions, then we can find a read other than $\mathbf{r}$ having the same $k$-mer profile. (i) Suppose that we have $r_1 \cdots r_{k-1} = r_{\ell-k+2} \cdots r_\ell$, then the read defined as $r_i r_{i+1} \cdots r_{\ell-k+2} \cdots r_{\ell-1} r_\ell r_k r_{k+1} \cdots r_{i-2} r_{i-1}$ will be distinct from $\mathbf{r}$ for some $i \in \{k, \ldots, \ell - k + 1\}$, and it shares the same $k$-mer profile. (ii) Let $\mathbf{r}$ be a read holding the second condition so we have $r_i \cdots r_{i+k-2} = r_j \cdots r_{j+k-2}$ and $r_g \cdots r_{g+k-2} = r_h \cdots r_{h+k-2}$ for some indices $1 \le i < g < j < h \le \ell$. Then, we can construct a different read having the same $k$-mer profile by interchanging the substrings $r_{i+k-1} \cdots r_{g-1}$ and $r_{j+k-1} \cdots r_{h-1}$. (iii) As the last condition, suppose that we have $r_i \cdots r_{i+k-2} = r_j \cdots r_{j+k-2} = r_h \cdots r_{h+k-2}$ for some indices $1 \le i < j < h \le \ell - k + 2$ such that substrings $r_{i+k-1} \cdots r_{j-1}$ and $r_{j+k-1} \cdots r_{h-1}$ are not the same so we can generate a new read with an equivalent $k$-mer profile by swapping these substrings.

Conversely, assume that there exists a read $\mathbf{q}$ that shares the same $k$-mer profile as $\mathbf{r}$ but doesn't meet any of the conditions outlined. Let $\mathbf{x}$ be the first $(k-1)$-mer in $\mathbf{r}$ where it appears more than once in $\mathbf{r}$, and let $i_1 \le i_2 \le \ldots \le i_n$ be the position indices where $\mathbf{x}$ occurs in $\mathbf{r}$, i.e. $\mathbf{x} = r_{i_h} \cdots r_{i_h+k-2}$ $\forall h \in \{1, \ldots, i_n\}$ and $c_{\mathbf{r}}(\mathbf{x}) = n$. It must satisfy $i_1 > 1$ or $i_n < \ell - k + 2$; otherwise, it violates the

first condition. By Lemma A.1, $\mathbf{r}$ and $\mathbf{q}$ must match up to the position index $i_n + k - 3$. Since $\mathbf{r}$ and $\mathbf{q}$ have the same $k$-mer profiles and $c_{\mathbf{r}}(r_{i_1-1} \cdots r_{i_1-k+1}) = c_{\mathbf{q}}(q_{i_1-1} \cdots q_{i_1-k+1}) = 1$, we can also write that $r_j = q_j$ for all $1 \le j \le i_1 + k - 2$ and we will show that reads, $\mathbf{r}$ and $\mathbf{q}$, must be the same up to the position index $i_n + k - 2$.

If $(k-1)$-mer, $\mathbf{x}$, occurs more than twice in $\mathbf{r}$ (i.e., $n > 2$), then either $i_{h+1} - i_h = 1$ or substrings $r_{i_h} \cdots r_{i_{h+1}}$ for each $h \in \{1, n-1\}$ must be the same due to the third condition. Thus, we obtain identical reads up to $i_n + k - 2$.

Now, we will consider the case in which $\mathbf{x}$ appears exactly twice in $\mathbf{r}$ (i.e., $n = 2$). If substring $r_{i_1+1} \cdots r_{i_n-1}$ consists only of $(k-1)$-mers occurring at most once in $\mathbf{r}$, then again, we achieve the equivalence of the reads up to index $i_n + k - 2$. Therefore, suppose that there exists a $(k-1)$-mer $\mathbf{y} \ne \mathbf{x}$ appearing more than once in $\mathbf{r}$ and let $j_1, \ldots, j_m$ be its position occurrence indices (i.e. $\mathbf{y} = r_{i_h} \cdots r_{i_h+k-2} \ \forall h \in \{1, \ldots, j_m\}$). Note that $j_m$ cannot be greater than $i_n$ since it violates the second condition, so we assume that all occurrences of $\mathbf{y}$ are within $r_{i_1+1} \cdots r_{i_n+k-3}$. By recursively applying the same strategies for the substring, $r_{j_1+1} \cdots r_{j_m-1}$, we can conclude that the reads must be the same up to $i_n + k - 2$.

Note that the remaining part of the read, $r_{i_n+k-1} \cdots r_\ell$, does not contain any $(k-1)$-mer appearing in $r_1 \cdots r_{i_n+k-2}$, so by applying the same strategy to the rest of the read, we can obtain that $\mathbf{r} = \mathbf{q}$. $\quad\square$

**Proposition A.4.** *Let $M_1 = (\aleph_\ell, d_{\mathcal{H}})$ and $M_2 = (\mathbb{N}^{|\Sigma^k|}, \|\cdot\|_1)$ be the metric spaces denoting the set of identifiable reads and their corresponding $k$-mer profiles equipped with edit and $\ell_1$ distances, respectively. The $k$-mer profile function, $c : M_1 \to M_2$, mapping given any read, $\mathbf{r}$, to its corresponding $k$-mer profile, $c_{\mathbf{r}} := c(\mathbf{r})$, is a Lipschitz equivalence, i.e. it satisfies*

$$\forall \mathbf{r}, \mathbf{q} \in \Sigma^\ell \quad \alpha_l d_{\mathcal{H}}(\mathbf{r}, \mathbf{q}) \le \|c_{\mathbf{r}} - c_{\mathbf{q}}\|_1 \le \alpha_u d_{\mathcal{H}}(\mathbf{r}, \mathbf{q}) \tag{7}$$

*for $\alpha_l = 1/\ell$ and $\alpha_u = k|\Sigma|^k$, so $M_1$ and $M_2$ are Lipschitz equivalent.*

*Proof.* By definition, we have $c_{\mathbf{r}}(\mathbf{x}) = \sum_{i=1}^{\ell-k+1} [r_i \cdots r_{i+k-1} = \mathbf{x}]$ where the symbol, $[\cdot]$, denotes the Iverson bracket. Then, we can write the upper bound as

$$\|c_{\mathbf{r}} - c_{\mathbf{q}}\|_1 = \sum_{\mathbf{x} \in \Sigma^k} \left| c_{\mathbf{r}}(\mathbf{x}) - c_{\mathbf{q}}(\mathbf{x}) \right|$$

$$= \sum_{\mathbf{x} \in \Sigma^k} \left| \sum_{i=1}^{\ell-k+1} [r_i \cdots r_{i+k-1} = \mathbf{x}] - \sum_{i=1}^{\ell-k+1} [q_i \cdots q_{i+k-1} = \mathbf{x}] \right|$$

$$= \sum_{\mathbf{x} \in \Sigma^k} \left| \sum_{i=1}^{\ell-k+1} \left( [r_i \cdots r_{i+k-1} = \mathbf{x}] - [q_i \cdots q_{i+k-1} = \mathbf{x}] \right) \right|$$

$$\le \sum_{\mathbf{x} \in \Sigma^k} \sum_{i=1}^{\ell-k+1} \left| [r_i \cdots r_{i+k-1} = \mathbf{x}] - [q_i \cdots q_{i+k-1} = \mathbf{x}] \right|$$

$$\le \sum_{\mathbf{x} \in \Sigma^k} \sum_{i=1}^{\ell-k+1} [r_i \cdots r_{i+k-1} \ne q_i \cdots q_{i+k-1}]$$

$$\le \sum_{\mathbf{x} \in \Sigma^k} \sum_{i=1}^{\ell-k+1} \sum_{n=0}^{k-1} [r_{i+n} \ne q_{i+n}]$$

$$\le \sum_{\mathbf{x} \in \Sigma^k} \sum_{i=1}^{\ell-k+1} k[r_i \ne q_i]$$

$$\le k \sum_{\mathbf{x} \in \Sigma^k} \sum_{l=1}^{L} [r_i \ne q_i]$$

$$= k|\Sigma|^k d_{\mathcal{H}}(\mathbf{r}, \mathbf{q})$$

For the lower bound, we know that $d_{\mathcal{H}}(\mathbf{r}, \mathbf{q})$ must be greater than $0$ if and only if $\mathbf{r}$ and $\mathbf{q}$ are different since $\mathbf{r}, \mathbf{q} \in \aleph_L$ (i.e. identifiable); otherwise we have $\|c_{\mathbf{r}} - c_{\mathbf{q}}\|_1 = d_{\mathcal{H}}(\mathbf{r}, \mathbf{q}) = 0$. For non-identical

reads, there must exist at least one $1 \leq i \leq \ell$ such that $r_i \neq q_i$, so it implies that there is an $k$-mer $\mathbf{x}' \in \Sigma^k$ such that $c_{\mathbf{r}}(\mathbf{x}') \neq c_{\mathbf{q}}(\mathbf{x}')$. Then we can write that

$$\frac{d_{\mathcal{H}}\left(\mathbf{r}, \mathbf{q}\right)}{\ell} \leq \|c_{\mathbf{r}} - c_{\mathbf{q}}\|_1$$

$\square$

## A.2   Experiments

For the evaluation of the models, we follow the same experimental set-up proposed by the recent genome foundation model [32], and we assess the performance of the models with respect to the quality of the number of detected clusters.

We suppose that the number of clusters (i.e., genomes) in the evaluation datasets is unknown, so we utilize the modified K-medoid algorithm[32] to infer the clusters. It requires a threshold value due to the initially unknown number of clusters so to address this, we use separate datasets derived from the same source as the target evaluation datasets. For each method, we first compute the cluster centroids using the ground-truth labels and the embeddings generated by the respective approaches. Then, we calculate the similarities between the centroid vector and the read embeddings within the same cluster. The threshold value is selected as the 70th percentile of these sorted similarity scores. After inferring the clusters by the modified K-medoid algorithm, the extracted cluster labels are then aligned with the ground truths labels by the *Hungarian* algorithm.

For the experiments, we use the publicly available datasets provided by Zhou et al. [32], which include a training set of 2 million non-overlapping DNA sequence pairs. For model evaluation, we work with 3 types of datasets, each having two variants, and we provide various statistics about them in Table 2. The datasets named as *Dataset* 0 are only used to detect the optimal threshold value for the K-medoid algorithm and the other variants (i.e., *Dataset* 5 and *Dataset* 6) are considered for the evaluation. We truncated sequences longer than $10,000$ bases, excluded those shorter than $2,500$ bases, and omitted species represented by fewer than 10 sequences.

Table 2: Statistics of the datasets used in the experimental evaluations.

| Dataset | Species | Sequences ($/\mathcal{R}/$) | Min. Seq. Length | Max. Seq. Length | Avg. Seq. Length |
|---------|---------|-----------|------------------|------------------|------------------|
| *Synthetic 0* | 200 | $20,000$ | $9,944$ | $10,000$ | $9,999$ |
| *Synthetic 5* | 323 | $37,278$ | $9,919$ | $10,000$ | $9,999$ |
| *Synthetic 6* | 249 | $28,206$ | $9,913$ | $10,000$ | $9,999$ |
| *Plant 0* | 108 | $10,800$ | $2,089$ | $20,000$ | $9,040$ |
| *Plant 5* | 323 | $10,800$ | $81$ | $20,000$ | $3,953$ |
| *Plant 6* | 374 | $111,395$ | $80$ | $20,000$ | $3,709$ |
| *Marine 0* | 326 | $32,600$ | $1971$ | $20,000$ | $9,600$ |
| *Marine 5* | 660 | $167,875$ | $83$ | $20,000$ | $4,755$ |
| *Marine 6* | 664 | $175,316$ | $80$ | $20,000$ | $4,840$ |

## A.3   Notation and Symbols

We provide a comprehensive list of all symbols used throughout the paper in Table 3. This table might serve as a reference to help readers easily follow the notation employed in our work.

Table 3: Symbols and their descriptions used throughout the paper.

| Symbol | Description |
|---|---|
| $\Sigma$ | Genetic alphabet |
| $\mathcal{R}$ | Set of reads |
| $\mathcal{E}(\cdot)$ | Embedding function |
| $\mathbf{r}$ & $\mathbf{q}$ | Read |
| $\ell$ | Read length |
| $k$ | Length of $k$-mers |
| $\Sigma^k$ | All $k$-mers of length $k$ |
| $y_{ij}$ | Binary variable showing if the pair is a positive sample |
| $p_{ij}$ | Success probability of a positive pair |
| $\lambda_{ij}, \lambda_{\mathbf{x},\mathbf{y}}$ | Rate of the Poison distribution |
| $\omega$ | Window size |
| $c_{\mathbf{r}}(\mathbf{x})$ | Number of occurrences of $\mathbf{x}$ in $\mathbf{r}$ |
| $o_{\mathbf{x},\mathbf{y}}(\mathbf{x})$ | Number of co-occurrences of $\mathbf{x}$ and $\mathbf{y}$ within a window of size $\omega$ |

