# OpenReview forum: "Revisiting K-mer Profile for Effective and Scalable Genome Representation Learning"
_NeurIPS.cc/2024/Conference — NeurIPS 2024 spotlight_

### Official Review · Reviewer_tV95 · 2024-06-18

**Soundness:** 3
**Presentation:** 3
**Contribution:** 2
**Rating:** 5
**Confidence:** 4

**Summary:**

This paper proposes a novel tokenization method for DNA sequence foundatnion model. According to the authors' experiments, their model has similar performances by comparing with other foundation models with larger scale, while their scalability looks better.

**Strengths:**

The tokenization process is interesting, and the paper contains theoricial proofs. It seems that their-TNF outperforms other methods in 1 out of 6 datasets.

**Weaknesses:**

Major:
1. Their proof majorly focus on arguing the problems of k-mers tokenization. However, some baselines like DNAHyena utilizes single nucleobase as tokens. Is it possible to justify the superority of the authors' method comparing with other tokenization approach (e.g., single nucleobase token, and BPE used by DNABERT-2).

2. According to their experiment result, it seems that their approach does not show comparable performances across all the datasets, which makes me hard to believe its superority. Moroever, some methods like HyenaDNA is only trained based on human genome, the testing dataset might be not suitable for benchmarking with these methods. Is it possible to select some genome overlapped with all of these methods? Also, it will be interesting if the authors can include NT [1] in the comparison.

3. In section 4.1, the authors utilize two datasets to analyze the effect of hyper-parameters, are there any reasons for not using other datasets?

4. The rest of benchmarked methods are all in Million level not billion level, and thus the superority of scaling it not very interesting. It will be helpful if the authors can discuss and try to scale their method to 1B or 7B level, which will be a breakthrough in this area. NT is 1B at max.

[1] https://github.com/instadeepai/nucleotide-transformer

Minor:

1. It seems that there is a block near line 176 and I cannot click it, is it a typo?

**Questions:**

Please see the weaknesses.

**Limitations:**

Please see the weaknesses, as well as their limitation section.

---

> ### Author Rebuttal · Authors · 2024-08-06
>
> We appreciate Reviewer tV95's thorough review and valuable comments. We address the reviewer's questions below:
>
> >**1. Their proof majorly focus on arguing the problems of k-mers tokenization. However, some baselines like DNAHyena utilizes single nucleobase as tokens. Is it possible to justify the superority of the authors' method comparing with other tokenization approach (e.g., single nucleobase token, and BPE used by DNABERT-2).**
>
> The work reported in our paper is partly motivated by the prevalent use of $k$-mer representations for, among other tasks, metagenomic binning. These representations have, however,  mainly been studied from an empirical perspective. With this paper, our aim is to also i) contribute to the theoretical understanding of $k$-mer representations and to ii) demonstrate that $k$-mers, together with simple and scalable model architectures, can provide a viable alternative to the more complex foundation models that have recently been proposed. Our experiments and comparisons have been designed with this focus in mind. While we fully agree that a rigorous comparison with other tokenization methods is interesting, we also believe that such a comparison is a bit out of scope for the present paper. However, it could indeed be relevant as part of future work. Please also see the first comment for *Reviewer xd8g*.
>
> >**2. According to their experiment result, it seems that their approach does not show comparable performances across all the datasets, which makes me hard to believe its superority. Moroever, some methods like HyenaDNA is only trained based on human genome, the testing dataset might be not suitable for benchmarking with these methods. Is it possible to select some genome overlapped with all of these methods? Also, it will be interesting if the authors can include NT [1] in the comparison.**
>
> As mentioned in our global response, our objective is to demonstrate that ML methods based on $k$-mers are surprisingly competitive with large foundational models and to provide a theoretical explanation for the $ k$-mer-based models. We are not claiming our methods are superior to other ML approaches. Instead, this paper supports the development of ML approaches leveraging $k$-mer-based representations to create more scalable metagenomic binning methods. As highlighted in the global response and by other reviewers, scalability is a crucial issue in metagenomic binning.
>
> In terms of accuracy results, we would also like to provide a bit of nuance to the comment that “their approach does not show comparable performances across all the datasets”. When focusing on the number of high-quality bins ($\geq 0.9$) that are recovered (which is typically the main focus in the metagenomic binning task), we see that ‘Ours-nonlinear’ outperforms DNABERT-S (as well as the other methods) on two data sets (Marine 5 and 6) and achieves comparable results on the remaining datasets.
>
> Lastly, we would also like to note that in our paper, we adopted the exact same experimental setup established by the DNABERT-S method, which is the latest state-of-the-art genome foundational model performing the metagenomic binning task. This setup is also endorsed by researchers involved with other leading genome foundational models. In this regard, we have run all the baseline methods as their authors suggested by following the established practices in the literature. See also the first comment for reviewer 8HNd.
>
> >**3. In section 4.1, the authors utilize two datasets to analyze the effect of hyper-parameters, are there any reasons for not using other datasets?**
>
> Because of the space limitations in the main paper, we chose only two datasets to show the impact of the hyper-parameters, but we provide the additional results in the attached file of the global response (Figures 1-6).
>
> >**4. The rest of benchmarked methods are all in Million level not billion level, and thus the superority of scaling it not very interesting. It will be helpful if the authors can discuss and try to scale their method to 1B or 7B level, which will be a breakthrough in this area. NT is 1B at max.**
>
> Our experiments demonstrate that we can achieve similar performance with $k$-mer-based representations using significantly smaller models. As you suggest, a very interesting experiment would be to create  $k$-mer-based models of the same size as existing foundational models to see if they perform better. We believe this is a promising line of research, and we hope this paper will motivate other researchers to explore this direction in future works.
>
> *[1] https://github.com/instadeepai/nucleotide-transformer*

---

> > ### Comment · Reviewer_tV95 · 2024-08-08
> > **Thanks for your responses**
> >
> > Thanks for your responses. However, I still have concerns about point 1 and point 4. If there is a better approach comparing with k-mer, why the analysis of k-mer design is more interesting and important? Is it possible to extend your theorical framework for analyszing other tokenization approach?
> >
> > Also, I think checking models with larger scale are required, which could help us understand the existance of scaling low and (possible) emergent abilities for DNA language model.

---

> > > ### Author Response · Authors · 2024-08-08
> > > **Thank you for the comments and questions**
> > >
> > > Thank you for the comments and questions, which we have tried to answer below.
> > >
> > > It is not our intention to claim that $k$-mer design is more interesting and important than all other tokenization methods. Instead, we see that k-mer tokenization is the most prevalent tokenization method for genome representations (not considering the advent of recent foundation models), but $k$-mer tokenization has mostly been explored from an empirical perspective. With our analysis, we aim to also contribute to the theoretical understanding of $k$-mer representations as well as demonstrate that k-mer representations can lead to the basis for designing simple or complex architectures that can be competitive and even better than the large genome foundation models but at a fraction of the cost.
> > >
> > > As for the question "*Is it possible to extend your theoretical framework for analyzing other tokenization approaches?*", we, unfortunately, do not see an immediate way to adapt the theorems and proofs to more general classes of tokenization methods. For example, the recursive nature of the BPE algorithm differs significantly from our setting. That being said, we agree that it could be interesting to perform similar types of theoretical analyses for other tokenization methods and our proof could potentially serve as inspiration for such analyses, but we respectfully feel that such an undertaking is outside the scope of the present paper.
> > >
> > > In terms of "*checking models with larger scale are required*" we are not quite sure what other types of models the reviewer have in mind. Our empirical analysis includes the latest SoA genome sequence foundation models (using the exact same empirical setup as used for DNABERT-S ) and our goal was to demonstrate that competitive results can be obtained at a fraction of the cost with much simpler architectures. Do you perhaps have particular architectures in mind that we might have missed and, if so, could you provide a reference to these?

---

> > > > ### Comment · Reviewer_tV95 · 2024-08-10
> > > > **Thanks for your responses**
> > > >
> > > > Thanks for your reply. I suggest reading some reveiw papers  (https://arxiv.org/html/2407.11435v1) to access different types of DNA language models, but like you said, many of them are not based on k-mer setting. Therefore, considering the limitation of current work, I will keep my score and stand in the side of acceptance.

---

> > > > > ### Author Response · Authors · 2024-08-12
> > > > > **Thank you for your thoughtful feedback**
> > > > >
> > > > > Thank you for your thoughtful feedback. Might we ask whether your support for acceptance indicates that you would consider raising your score to increase the chance of having the paper accepted?

---

### Official Review · Reviewer_8HNd · 2024-06-29

**Soundness:** 3
**Presentation:** 3
**Contribution:** 3
**Rating:** 5
**Confidence:** 3

**Summary:**

The paper explores the use of k-mer profiles in genome representation for metagenomic binning. The authors propose a new, scalable model that leverages k-mer profiles to represent DNA fragments efficiently. This model is theoretically grounded and empirically validated against state-of-the-art genome foundation models.

**Strengths:**

1. **Theoretical Contributions**: The paper broadens the theoretical understanding of k-mer. By establishing theoretical bounds and providing a detailed analysis of DNA fragment identifiability, it contributes valuable insights that can guide future research and applications in genome analysis.

2. **Scalability and Computational Efficiency**: The proposed model addresses the computational demands associated with large genomic datasets. By demonstrating that their k-mer-based model requires significantly fewer resources compared to more complex genome foundation models, the paper appeals to practical needs in genomic research, particularly in resource-limited scenarios.

3. **Empirical Validation**: Through extensive empirical testing, the paper substantiates the theoretical claims by showing that the k-mer-based model performs comparably to genome foundation models in metagenomic binning tasks. This not only validates the model but also emphasizes its practicality for real-world applications.

**Weaknesses:**

1. **Comparison Basis**: The paper positions k-mer profiles as a competing approach against DNA language models (LMs). This comparison is somewhat misleading as k-mer is fundamentally a tokenization method (i.e. one initial step of DNA FM), and a more appropriate comparison would be between different tokenization strategies (k-mer vs. BPE) rather than against the entirety of DNA LMs.

2. **Experimental Clarity**: The experimental setup lacks detailed discussion, particularly on how the DNA foundation models are integrated and utilized in the experiments. There is a concern about potential unfair comparisons, especially if the study only uses the embeddings from DNA LMs for classification without fine-tuning the full models, which is the standard practice for such models. This could lead to skewed results favoring the proposed method.

3. **Technical Details and Minor Errors**: The paper has several minor but notable issues, such as the lack of clarity in the definition of the loss function at line 199 and missing labels for equations post Equation 3. These issues, while minor, could impede the understanding and reproducibility of the research.

**Questions:**

See W2, W3.

---

> ### Author Rebuttal · Authors · 2024-08-06
>
> We appreciate Reviewer 8HNd 's assessment and valuable comments. We address the reviewer’s questions in detail below:
>
> **1. Comparison Basis:**
>
> In the literature, $k$-mers are widely used as feature vectors, but their effectiveness has not been examined well and is often attributed solely to practical reasons, such as the fixed-length input vectors. To address this, we reexamined $k$-mer features to understand why they are effective, and we explored the theoretical connection between DNA sequences and their $k$-mer profiles.
>
> Although the $k$-mers are a tokenization method, we demonstrated with simple $k$-mer based models that it is still possible to have comparable performances to large genome foundational models without requiring extensive computational resources. Our main motivation was to show that $k$-mer profiles remain powerful features (motivated by our theoretical analysis), and $k$-mer based models can achieve comparable or even superior results in the metagenomic binning task compared to these large foundational models. With advancements in sequencing technologies, large datasets have become more accessible, making the scalability of models a critical consideration. We believe our analysis, grounded in theoretical principles, can spark the development of novel $k$-mer-based approaches for obtaining more scalable and efficient methods, potentially playing a significant role in future research.
>
>
> **2. Experimental Clarity:**
>
> In our paper, we adopted the same experimental setup established by the DNABERT-S method [3], which is the latest state-of-the-art genome foundational model performing the metagenomic binning task. This setup is also endorsed by researchers involved with other leading genome foundational models [1, 2]. In this regard, we have run all the baseline methods as their authors suggested by following the established practices in the literature.
>
> We also utilized the publicly available datasets provided by the authors of DNABERT-S [4], and we shared all source codes for our proposed models to ensure that our results are easily reproducible. By adhering to established standards in the field, we maintain consistency and reliability in our experimental settings.
>
> *[1] Ji, Yanrong, et al. "DNABERT: pre-trained Bidirectional Encoder Representations from Transformers model for DNA-language in genome." Bioinformatics 37.15 (2021): 2112-2120.*
>
> *[2] Zhou, Zhihan, et al. "Dnabert-2: Efficient foundation model and benchmark for multi-species genome." arXiv preprint arXiv:2306.15006 (2023).*
>
> *[3] Zhou, Zhihan, et al. "Dnabert-s: Learning species-aware dna embedding with genome foundation models." ArXiv (2024).*
>
> *[4] https://github.com/MAGICS-LAB/DNABERT_S*
>
>
>
> **3. Technical Details and Minor Errors:**
>
> We appreciate the reviewer for pointing out the notational mistakes, and we will correct all errors in the final version of the paper.

---

> > ### Comment · Reviewer_8HNd · 2024-08-08
> > **Response to Rebuttal**
> >
> > I've carefully read through the rebuttal and the attached PDF file. I appreciate the authors' efforts in addressing the weaknesses I identified.
> >
> > I have increased my score as the authors have adequately addressed my concerns.

---

> > > ### Author Response · Authors · 2024-08-08
> > > **Response**
> > >
> > > We appreciate your thoughtful feedback and are glad that our revisions have addressed your concerns.

---

### Official Review · Reviewer_e86P · 2024-07-11

**Soundness:** 3
**Presentation:** 4
**Contribution:** 4
**Rating:** 8
**Confidence:** 4

**Summary:**

The authors describe an embedding approach for binning metagenomic reads via a k-mer approach. The authors motivate the need for scalable solutions for metagenomic analysis and provide theoretical motivation for k-mer based approaches, which have value on their own. Then they motivate a strategy to learn non-linear embedding of short, correlated k-mers for binning purposes. They compare their approach to several large language models that have been used for that purposes and while having much higher complexity and foodprint, these do not provide better performance on commonly used standard benchmarking data.

**Strengths:**

* very well written manuscript, nicely incorporating algorithmic and learning approach for a relevant application
•	Impressive results (considering the scalability of the solution and the importance of scalability for the problem) in a well chosen benchmark
•	Good theoretical foundation of the provided solution (which has value beyond the proposed solution)

**Weaknesses:**

* The authors may want to acknowledge a little more in depth that there are also excellent algorithmic solutions for metagenomic binning that are directly evaluated on the CAMI2 data (in the CAMI2 publication).
* Related, there is also no comparative evaluation to algorithmic approaches available (although very good ones to other machine learning based approaches).

**Questions:**

* Could the authors reconfirm that there is no overlap between the training data and the CAMI2 testing data? How was this ensured?
* I understand the reasoning to not show errors bars, but could the authors (in the appendix) still provide the data for completeness reasons?
* I am not sure this is feasible and this is really a discretionary question out of curiousity (and does not need to be answered for changing my mind on acceptance). Can the authors establish a conceptual relation of their embeddings to the probabilistic data structures (most recent example: https://genome.cshlp.org/content/early/2024/06/17/gr.278623.123.abstract) that are commonly used in algorithmic binning?
* Can the authors compare to algorithmic approaches on CAMI2 data to give a ballpark orientation of the performance of their learning based approach?

**Limitations:**

Limitations are well addressed in the manuscript

---

> ### Author Rebuttal · Authors · 2024-08-06
>
> We are grateful to Reviewer e86P for the constructive review and helpful recommendations. We address Reviewer e86P’s concerns in detail below.
>
> > **The authors may want to acknowledge a little more in depth that there are also excellent algorithmic solutions for metagenomic binning that are directly evaluated on the CAMI2 data (in the CAMI2 publication).**
>
> Thanks for reminding us of the inclusion of CAMI2. We will discuss it in the paper.
>
> > **There is also no comparative evaluation to algorithmic approaches available**
>
> We completely agree that algorithmic approaches can be highly competitive with machine learning (ML) methods. However, we believe that such a comparison is not necessary for this particular study. Our objective is to demonstrate that ML methods based on k-mers are surprisingly competitive with large foundational models and to provide a theoretical explanation for this phenomenon. We are not asserting that our methods are superior to other ML or algorithmic approaches. Instead, we view this paper as evidence supporting the development of ML approaches that leverage $k$-mer-based representations to design more scalable meta-genomic binning methods.
>
> > **Could the authors reconfirm that there is no overlap between the training data and the CAMI2 testing data? How was this ensured?**
>
> We utilized the publicly available datasets provided by the authors of the genome foundational model, DNABERT-S, and the authors have stated that the training and testing sets do not overlap.
>
> > **I understand the reasoning to not show errors bars, but could the authors (in the appendix) still provide the data for completeness reasons?**
>
> We provide the detailed results for five runs of the non-linear model in the attachment of the global comment (Tables 1 & 2), and we will also include them in the appendix of the final version of the paper.
>
> > **I am not sure this is feasible and this is really a discretionary question out of curiousity (and does not need to be answered for changing my mind on acceptance). Can the authors establish a conceptual relation of their embeddings to the probabilistic data structures**
>
> Thank you for the highly relevant question. The paper referenced above together with other state-of-the-art taxonomic classifiers such as [1], rely on or incorporate $k$-mer based representations for querying reference databases. The theoretical results in our paper can thus provide support for these approaches, but it would be interesting to investigate whether our results could be further developed to more specifically address taxonomic classification.  We have acknowledged this relation in the paper and included the two references.
>
> *[1] Wei et al: Kmcp: accurate metagenomic profiling of both prokaryotic and viral populations by pseudo-mapping. Bioinformatics, 39(1), 2023.*
>
> > **Can the authors compare to algorithmic approaches on CAMI2 data to give a ballpark orientation of the performance of their learning based approach?**
>
> As we have previously stated, while we agree that this comparison would be quite interesting, we believe it is beyond the scope of this paper. Our focus is to demonstrate and analyze why $k$-mer-based ML methods are competitive with large foundational models for the metagenomic binning task.

---

> > ### Comment · Reviewer_e86P · 2024-08-09
> >
> > Thank you for clarifying.
> > As my score was already optimistic in the original review and I only had minor/discretional questions, I am not adjusting here.

---

### Official Review · Reviewer_xd8g · 2024-07-12

**Soundness:** 3
**Presentation:** 3
**Contribution:** 2
**Rating:** 5
**Confidence:** 3

**Summary:**

The authors provides a theoretical analysis for k-mer-based representation of genomes and then propose a lightweight and scalable model for performing metagenomic binning at the genome read level, relying on the k-mer compositions of the DNA fragments and achieve pretty good results.

**Strengths:**

1. The logical is exceptionally clear and well-structured.

2. The issues examined in this paper are interesting and relevant.

**Weaknesses:**

1. Limited comparison with the current methods. Zhang[1] has propose a novel k-mer natural vector for representing biological sequences with the consideration of positional information in sequence. What is your strengths compared with the novel k-mer natural vector. Could you make comparison with it in the task of Metagenomics Binning. In addition, there are various pseudo features extracted from biological sequences directly [2], why not compare with them in terms of performance.

2. Limited experimental results. The article just shows the results just on one task of Metagenomics Binning, which may not fully demonstrate the effectiveness of k-mer. If possible, please design more downstream tasks.

3. The poor framework. We all know k-mer feature can work, but how to design a framework to its strength. As shown in the Figure 3, DNABERT-S outperforms ‘Ours-linear’ and ‘Ours-nonlinear’ in all datasets. In addition, METABAT2, VAMP and SEMIBIN2 are the common binning algorithm, why not compare with them to highlight the effectiveness of “Ours-linear” and “Ours-nonlinear”.

4. Lack of guidance from theory to experiment. The proof of Theorem 3.1 is a wonderful process, but how to use it to guide the experiment, how to select a suitable k according the special dataset. And if we consider the position of k-mers in sequences, then we can assure that any sequences are identifiable unless they are exactly the same.

5. Lack more detailed explanation for the effectiveness of k-mers. The DNABERT family builds a vocabulary based on k-mers and then uses a nonlinear network to extract the embedding for each tokens, which is also a nonlinear representation, but what is the difference between this and "Ours nonlinear". Apart from the dimensional difference, where is the disadvantage.

Reference:

[1] Zhang, YuYan, Jia Wen, and Stephen S-T. Yau. "Phylogenetic analysis of protein sequences based on a novel k-mer natural vector method." Genomics 111.6 (2019): 1298-1305.

[2] Pse-in-One: a web server for generating various modes of pseudo components of DNA, RNA, and protein sequences B Liu, F Liu, X Wang, J Chen, L Fang, KC Chou - Nucleic acids research, 2015

**Questions:**

See Weekness

**Limitations:**

Yes, they Have.

---

> ### Author Rebuttal · Authors · 2024-08-06
>
> We are thankful to Reviewer xd8g for the thoughtful feedback and suggestions. We address the reviewer's questions in detail below:
>
> **1. Limited comparison with the current methods.**
>
> Thank you for your helpful reference, which we have added to the conclusions/future work section.  We agree that exploring alternative k-mer-based representations [1] and/or pseudo-features [2] could indeed be interesting in relation to metagenomic binning. However, our aim is not to determine the best $k$-mer or pseudo-feature representation (or tokenization in general). Rather, we aim to i) contribute to the theoretical understanding of standard $k$-mer-based representations used in the literature and to ii) demonstrate how simple $k$-mer-based model architectures can provide competitive accuracy results compared to recently proposed foundation models, but using only a fraction of the number of model parameters. We thus believe that even though a comparison with other tokenization methods could indeed be interesting, we also feel that it is a bit out of scope for the present paper, but could indeed be a relevant topic for future work. Please see also the first comment for *Reviewer tV95*.
>
> **2. Limited experimental results.**
>
> We acknowledge the importance of evaluating a model across a broad range of tasks. However, metagenomics binning is a complex and challenging task on its own. It is an emerging field attracting significant attention and is a primary approach for identifying new species [3,4]. Consequently, any advancement in this area is highly relevant. Given the complexity of this study and the application domain, we have focused exclusively on the metagenomics binning problem. We plan to extend our analysis to include additional downstream tasks in future work.
>
> *[3] Dongwan D. Kang, Feng Li, Edward Kirton, Ashleigh Thomas, Rob Egan, Hong An, and381 Zhong Wang. MetaBAT 2: An adaptive binning algorithm for robust and efficient genome382 reconstruction from metagenome assemblies. PeerJ, 7:e7359, 2019.*
>
> *[4] Singleton et al. (2024, jun. 27). Microflora Danica: the atlas of Danish environmental microbiomes. https://doi.org/10.1101/2024.06.27.600767*
>
>
> **3. The poor framework.**
>
> As noted for Question 1 above, the aim of this paper is to i) contribute to the theoretical understanding of standard $k$-mer-based representations used in the literature and to ii) demonstrate how simple $k$-mer-based model architectures can provide competitive accuracy results compared to recently proposed foundation models, but using only a fraction of the number of model parameters. Our goal is, therefore, not to propose a novel competitive binning method, which (we agree) should otherwise have been compared to state-of-the-art binners like VAMB, METABAT2, or SemiBin2. That being said, our non-linear model is inspired by SemiBin2 with a similar neural network architecture for deriving embeddings from $k$-mers. Additionally, we employ the same clustering algorithm as DNABERT-2 to ensure that the clustering algorithm itself does not influence the comparison, and, as discussed in the global response, we are using the exact same experimental setup as in the DNABERT-S paper.
>
> Lastly, we would also like to provide a bit of nuance to the comment that “DNABERT-S outperforms ‘Ours-linear’ and ‘Ours-nonlinear’ in all datasets.” When focusing on the number of high-quality bins ($\geq 0.9$) that are recovered (which is typically the main focus in the metagenomic binning task), we see that ‘Ours-nonlinear’ outperforms DNABERT-S on two data sets (Marine 5 and 6) and achieves comparable results on the remaining datasets.
>
>
> **4. Lack of guidance from theory to experiment.**
>
> In this paper, we explored the relationship between reads (i.e., small DNA fragments) and their $k$-mer profiles to understand how $k$-mers can approximate distances within the read spaces. We introduced the concept of "identifiability" for the reads and demonstrated that these sequences can be perfectly reconstructed from their given $k$-mer profiles. It is also straightforward to see that setting the $k$ value to the length of the reads makes all sequences identifiable. However, finding the smallest $k$ value that makes all (or a certain percentage of) reads identifiable is challenging.
>
> Our primary interest is to obtain similar read representations in a latent space for the reads belonging to the same genome/species so that clustering methods can easily distinguish reads with respect to their species. However, reads seeming dissimilar might belong to the same species, so we use linear and non-linear models to learn the underlying patterns and properties among these reads, enabling them to be represented closely within a latent space. In this regard, we relax the identifiability condition to give more flexibility to the models. Our empirical findings indicate that using around $k=4$ provides optimal results for the metagenomic binning task (please see Figure 4 in the main paper and Figures 1-3 in the attached pdf file of the global comment).
>
> **5. Lack more detailed explanation for the effectiveness of k-mers. What is the difference between this and "Ours nonlinear". Apart from the dimensional difference, where is the disadvantage.**
>
> It is correct that the original DNABERT model is based on $k$-mers, but the more recent DNABERT versions, which are used for our comparisons (DNABERT-2 and DNABERT-S), rely on byte pair encoding as tokenization. For DNABERT-2, the authors note that DNABERT-2 has 21x fewer parameters than DNABERT, and since our experiments were designed to demonstrate that comparative performance can be obtained with simpler and more scalable model architectures, the DNABERT method was not included in the comparison.  The DNABERT-S method is also the recent state-of-the-art genome foundation model considering the metagenomic binning task. In this regard, we have included the DNABERT-2 and DNABERT-S approaches in our experimental analysis.

---

> > ### Comment · Reviewer_xd8g · 2024-08-13
> > **Thank you for your reply and I will keep my positive comments.**
> >
> > Thank you for your reply and I will keep my positive comments.

---

### Author Rebuttal · Authors · 2024-08-06

We sincerely thank all the reviewers for their valuable feedback and thoughtful insights, which we believe will greatly enhance the quality and clarity of the final version of the paper. While we have addressed each reviewer's questions in their respective rebuttal sections, we would like to provide further clarification on the following main points:

**1. Motivation:**

The main purpose of the paper is to show how $k$-mer-based representation, combined with a simple non-linear model and chosen appropriate metric spaces, can compete with or even surpass the performance of genome foundational models in the metagenomic binning task. The presented surprising results demonstrate that the community needs to better understand why $k$-mers are effective features because, despite their widespread use in various DNA sequence analysis tasks, the reasons behind their effectiveness and their potential as alternative representations of DNA sequences have not been thoroughly studied.

Therefore, our paper focuses on the relationship between DNA sequences and their corresponding $k$-mer profiles, and we provide detailed theoretical analysis to comprehend how $k$-mers can approximate distances and similarities in the DNA sequence spaces.  We believe our theoretical and experimental analysis will help researchers understand why the $k$-mer profiles are effective representations, as well as inspire other researchers to design better and more scalable k-mer-based models for metagenomic binning tasks.


**2. A scalable approach:**

As we argue in the paper and other reviewers have noticed, scalability is an important issue in metagenomic binning tasks.  With advancements in sequencing technologies, large-scale genome datasets have become more prevalent [1]. However, addressing these datasets with resource-intensive models is challenging. Therefore, we aimed to demonstrate that one can have more lightweight models relying on $k$-mers that balance performance and computational requirements. Our experimental evaluations showed that the proposed architecture can achieve performance comparable to recent large genomic foundational models while using a number of parameters which are several orders of magnitudes smaller.


**3. Experimental setup and results:**

In the paper, we used exactly the same experimental setup proposed by the DNABERT-S method [4]. It is the most recent state-of-the-art genome foundational model for the metagenomic binning task, which was also suggested by researchers, including the authors of the other foundational models [2,3]. Similarly, for the training and testing sets, we utilized the publicly available datasets provided by the authors [5].

We have made all the source codes for our proposed models available so all results can be easily reproducible. In this regard, our experimental settings follow the established standards in the field, ensuring consistency and reliability. We believe that this transparency and adherence to recognized methods will facilitate further research and validation of our approach. The additional experimental results can be found in the attached file.

*[1] Singleton, Caitlin M., et al. "Microflora Danica: the atlas of Danish environmental microbiomes." bioRxiv (2024): 2024-06.*

*[2] Ji, Yanrong, et al. "DNABERT: pre-trained Bidirectional Encoder Representations from Transformers model for DNA-language in genome." Bioinformatics 37.15 (2021): 2112-2120.*

*[3] Zhou, Zhihan, et al. "Dnabert-2: Efficient foundation model and benchmark for multi-species genome." arXiv preprint arXiv:2306.15006 (2023).*

*[4] Zhou, Zhihan, et al. "Dnabert-s: Learning species-aware dna embedding with genome foundation models." ArXiv (2024).*

*[5] https://github.com/MAGICS-LAB/DNABERT_S*

---

### Decision · Program_Chairs · 2024-09-25

**Decision:**

Accept (spotlight)

**Comment:**

This paper presents a k-mer binning algorithm using language models, which is an emerging area. The paper appears to be well written and all reviewers are borderline accept or above, with one strong accept. From a quick look myself, the theoretical result and experimental results aren't per say strong/extensive, but appear to be well-presented/written. This clearly is an accept, and a spotlight is appropriate.